# An Objective and Repeatable Sac Isolation Technique for Comparing Biomechanical Metrics in Abdominal Aortic Aneurysms

**DOI:** 10.3390/bioengineering9110601

**Published:** 2022-10-22

**Authors:** Timothy K. Chung, Pete H. Gueldner, Trevor M. Kickliter, Nathan L. Liang, David A. Vorp

**Affiliations:** 1Department of Bioengineering, University of Pittsburgh, Pittsburgh, PA 15260, USA; 2Department of Mechanical Engineering and Materials Science, University of Pittsburgh, Pittsburgh, PA 15261, USA; 3Department of Surgery, University of Pittsburgh, Pittsburgh, PA 15213, USA; 4Division of Vascular Surgery, University of Pittsburgh Medical Center, Pittsburgh, PA 15213, USA; 5McGowan Institute for Regenerative Medicine, University of Pittsburgh, Pittsburgh, PA 15219, USA; 6Department of Chemical and Petroleum Engineering, University of Pittsburgh, Pittsburgh, PA 15261, USA; 7Department of Cardiothoracic Surgery, University of Pittsburgh, Pittsburgh, PA 15213, USA; 8Clinical and Translational Sciences Institute, University of Pittsburgh, Pittsburgh, PA 15213, USA

**Keywords:** abdominal aortic aneurysm, sac isolation, finite element analysis, biomechanics

## Abstract

(1) Abdominal aortic aneurysm (AAA) biomechanics-based metrics often reported may be over/under-estimated by including non-aneurysmal regions in the analyses, which is typical, rather than isolating the dilated sac region. We demonstrate the utility of a novel sac-isolation algorithm by comparing peak/mean wall stress (PWS, MWS), with/without sac isolation, for AAA that were categorized as stable or unstable in 245 patient CT image sets. (2) 245 patient computed tomography images were collected, segmented, meshed, and had subsequent finite element analysis performed in preparation of our novel sac isolation technique. Sac isolation was initiated by rotating 3D surfaces incrementally, extracting 2D projections, curve fitting a Fourier series, and taking the local extrema as superior/inferior boundaries for the aneurysmal sac. The PWS/MWS were compared pairwise using the entire aneurysm and the isolated sac alone. (3) MWS, not PWS, was significantly different between the sac alone and the entire aneurysm. We found no statistically significant difference in wall stress measures between stable (*n* = 222) and unstable (*n* = 23) groups using the entire aneurysm. However, using sac-isolation, PWS (24.6 ± 7.06 vs. 20.5 ± 8.04 N/cm^2^; *p* = 0.003) and MWS (12.0 ± 3.63 vs. 10.5 ± 4.11 N/cm^2^; *p* = 0.022) were both significantly higher in unstable vs. stable groups. (4) Our results suggest that evaluating only the AAA sac can influence wall stress metrics and may reveal differences in stable and unstable groups of aneurysms that may not otherwise be detected when the entire aneurysm is used.

## 1. Introduction

An abdominal aortic aneurysm (AAA) is a local dilatation of the abdominal aorta beyond 50% of its original size. If left untreated, an aneurysm can rupture, an event associated with a mortality rate of 90%, making it the 13th leading cause of death in the United States [1]. Current guidelines [2] for clinical intervention are to perform surgery when the AAA reaches 5.5 cm for men or 5.0 cm for women, under the premise that the risk of surgery at smaller diameters is greater than the risk of rupture. However, it has been reported that 13% [3] to 23.4% [4,5] of smaller-sized aneurysms still rupture, suggesting that there would be a benefit to establishing a more accurate decision tool for clinical intervention. AAA rupture is primarily a local biomechanical event where the transmural AAA wall stress exceeds the wall tissue strength. The rupture potential index (RPI) was introduced [6] to quantitatively express the likelihood of rupture occurring at a singular location on an aneurysm, and was developed using both experimental testing and computational simulations derived from experimentally validated material models [7,8]. The one-dimensional maximum diameter criterion has remained a poor predictive metric due to the risk of small aneurysm rupture [3,4,5]. The RPI, a ratio of wall stress to strength, was developed to provide a noninvasive methodology to quantify risk (i.e., identify locations where the wall stress to strength is elevated). A commercially available software exists, A4clinicsRE (VASCOPS GmbH, Graz, Austria) that uses a semi-automated approach to perform image segmentation, 3D surface reconstruction, and stress analysis. VASCOPS uses the three-dimensional geometry to assess risk of rupture, it is then related to the maximum diameter based on equivalent diameter risk [9]. The biomechanics-based prediction of aneurysm rupture risk (BIOPARR) is a free software package that is semi-automated and utilizes the RPI to perform risk analysis of AAA [10].

The most commonly reported metrics of AAA transmural wall stress are the peak wall stress (PWS) and the mean wall stress (MWS). These are typically extracted from finite element analysis (FEA), but their exact quantitative definition has varied among reports in the literature. For example, Speelman et al. suggested using the 99th percentile for PWS, as opposed to the true singular maximum, or “peak”, stress value, as it is more reproducible and less susceptible to edge effects and other artifacts [11]. However, PWS is often misused or misinterpreted. While it represents the maximum stress acting for the entire aneurysm, it does not necessarily indicate the site of rupture since the local transmural wall stress is only half the consideration, i.e., the corresponding local tissue strength must also be considered. Nonetheless, a statistical difference in PWS has been seen when comparing symptomatic/ruptured AAA to elective repair groups [12]. Most reports of MWS in AAA used the average of the calculated wall stresses at all nodes of the aneurysm wall after removing boundary nodes, which can erroneously inflate stress values due to edge effects. However, inclusion of non-aneurysmal portions of the aorta in FEA analysis may still introduce error in calculating MWS.

The primary and significant source of variability in AAA biomechanical metrics is the definition of the spatial region over which the metrics are considered. For example, some reports of AAA biomechanics defined their metrics over a region that included the iliac arteries up to and including the superior mesenteric arteries [6,12,13,14,15,16,17,18,19], while some have used only the region from the aortic bifurcation point up to the most distal renal artery. These different definitions do not allow for meaningful comparisons of some biomechanical and morphological metrics between AAA and across studies. For example, the extraneous visceral and iliac arteries tend to artificially lower the MWS. Perhaps as a result, this metric has been largely overlooked in AAA biomechanics because it has not been shown to date to be useful for comparisons between relevant groups. We believe that if the region of interest is carefully defined so that it is repeatable and reproducible, a biomechanics-based metric (such as MWS or PWS) could be used as a biomarker to permit comparison across AAA populations or groups.

In this paper, we propose an algorithmic means for isolating the aneurysmal sac in patient-specific 3D reconstructed models for computational analysis of AAA to more accurately and repeatably report biomechanical metrics that are more suitable for comparing groups. We demonstrate the usefulness of the algorithm by comparing PWS and MWS between the isolated aneurysm sac and the entire aneurysm, as well as to compare both metrics for isolated sacs between stable and unstable (rupture and intervention) AAA.

## 2. Materials and Methods

### 2.1. Clinical Dataset and Imaging Repository: De-Identified Sets of Clinical and Imaging

Data from a total of 778 patients within the University of Pittsburgh Medical Center system diagnosed with AAA were delivered via Globus cloud services (Argonne National Laboratory, Chicago, IL, USA) by the Health Record Research Request, a service of the Department of Biomedical Informatics at the University of Pittsburgh. Of these, CT angiography was not of sufficient quality for our purposes in 393 cases. Aneurysms with no detectable intraluminal thrombus (ILT) were excluded from this study to account for that common feature in the analyses, which left 245 cases analyzed in the current study.

### 2.2. Image Segmentation of AAA from CT: Image Segmentation from CT Angiography

DICOM data was performed using a semi-automated approach with a U-NET convolutional neural network and custom MATLAB code (Mathworks, Natwick, MA, USA) [18,19] (Figure 1). The U-NET was trained using Amazon Web Service’s (Amazon, Seattle, WA, USA) Elastic Compute Cloud and local workstations using multiple graphics processing units [20,21]. All image sets were input into the U-NET and were manually verified. The output for each segmented axial slice was a delineation of regions defining the lumen, wall, and ILT that were converted into point clouds in 3D Cartesian coordinates. The point clouds were scaled using the pixel spacing retrieved from the DICOM metadata, and the Z-spacing was scaled using the intervals between the image patient positions. The point clouds for the lumen and wall were converted into an initial mesh by computing the vertex normal and performing Poisson surface reconstruction [22] with a neighborhood of 15 vertices using an automated script. The meshes were smoothed using a surface preserving Laplacian smoothing technique [23]. The initial mesh surfaces of the lumen and wall served as the base building block of the finite element model.

### 2.3. Finite Element Analysis

The FEA used in this study followed a well-established general process [7,24,25] and incorporated previously published, experimentally measured material properties of the aneurysm wall and ILT [6,7,8,25]. After the segmentation of the aorta from the most proximal renal artery to the iliac bifurcation (the “entire aneurysm”), preliminary meshes for the lumen and wall were converted into polysurfaces. These were then imported into the meshing software ANSYS ICEM (Ansys 2019 R3, Inc., Canonsburg, PA, USA), where the wall was meshed with 2D shell elements (assigned S3R for Abaqus) and the ILT was meshed with 3D tetrahedral elements (assigned C3D4 for Abaqus). An established, validated hyperelastic isotropic material model was used for the ILT [24], and an established anisotropic model was used for the AAA wall material [7] using a user-defined function. The isotropic ILT model uses a material model consisting of two parameters c_1_ = 2.6 N/cm^2^, and c_2_ = 2.6 N/cm^2^ [8,24,25]. The biaxial behavior was previously modeled and descried by Vande Geest et al. 2008 [8] where the strain energy assumption is defined by the following form (1):(1)$$W=b_{o}\left(e^{\frac{1}{2}b_{1}E_{\theta \theta }^{2}\;}+e^{\frac{1}{2}b_{2}E_{LL}^{2}\;}+e^{b_{3}E_{\theta \theta }E_{LL}\;}+e^{\frac{1}{2}b_{4}E_{\theta L}^{2}\;}+e^{b_{5}E_{\theta \theta }E_{\theta L}\;}+e^{b_{6}E_{LL}E_{\theta L}\;}-6\right)\;\;$$
where *b_*0*_, b_*1*_, b_*2*_, and b_*3*_* are 0.14, 477.0, 416.4, and 408.3 kPa, respectively [7] The strain energy terms involving the constants *b_*4*_*, characterize material shear and *b_*5*_*, and *b_*6*_* the shear-normal behaviors (included for completeness, but is not required for fitting experimental stress–strain data) [8]. A uniform AAA wall thickness of 1.9 mm was assumed [23,24] and boundary conditions were the constraint of the proximal end of the AAA model in the x, y, and z directions and the application of an ideal systolic pressure of 120 mmHg was applied as a distributed load normal to the surface of the lumen elements. In a previous study by Chung et al. [15], zero-pressure geometry AAA were reconstructed and stress analysis performed using ideal systolic blood pressure (120 mmHg). The deformed geometry was then subject to an additional 120 mmHg, simulating a medical image taken at ideal systolic pressure. It was found that there was a 3% increase in wall stress when comparing a zero-pressure heterogeneous model and a pre-deformed (i.e., not at a zero-pressure state) homogeneous model. Therefore, it was reasonable to perform stress analysis from medical images using the surfaces as the initial condition (even though the initial pre-stress or zero-pressure geometry is unknown).

All FEA simulations were performed in Abaqus Standard—implicit mode (Dassault Systemés, Waltham, MA, USA) with Microsoft Visual Studio 2017 (Microsoft Inc., Redmond, WA, USA) and Intel Fortran Compiler (Intel Inc., Santa Clara, CA, USA) to accommodate the user-defined function. The geometries were visualized using Tecplot (Tecplot Inc., Bellevue, WA, USA). After each simulation was completed, a custom MATLAB post-processing script was run on each case to extract and visualize von Mises wall stress for each node of the aneurysm wall model.

### 2.4. Mesh Independence Study and Mesh Quality Checks

A mesh independence study was performed to determine the minimum number of elements that represented a converged solution (reporting peak wall stress within the aneurysm sac). The global mesh seed size and density targets were perturbed incrementally, and the simulations were run using Abaqus Standard (implicit mode). The peak wall stresses were recorded and plotted against the number of elements. A spline interpolant was used to fit the data for presentation and the number of target number of elements was chosen based on the mesh density parameters. The mesh quality was also assessed in parallel when performing the meshing (in ICEM) to ensure that there were no non-manifold edges, zero volume elements, and high aspect ratio surface elements.

### 2.5. Algorithmic Isolation of the Aneurysm Sac

Sac isolation was performed using the post-processed 3D reconstructed wall surface of the aneurysm and the von Mises wall stresses calculated for each node. Each AAA model was translated spatially so that its centroid was located at the origin (0, 0, 0) in Cartesian coordinates. The Z-axis corresponding to the craniocaudal axis in the supine position was used as a reference axis for rotation of the 3D reconstructed aneurysm to produce a number of projected boundaries. The rotational degrees corresponded directly to the number of planes taken to determine the superior and inferior boundaries of the sac (e.g., there are 360 rotational increments for 1°). The number of angular rotational increments, *θ_z_*, is directly related to the number of projected boundaries from which to identify the sac region (i.e., the ratio of the total rotation of 360° and the rotational increment *θ_z_* is equivalent to the number of projected boundaries). The 3D nodal coordinates were transformed at each increment of rotation using a rotational matrix as follows (2):(2)$$\left[\begin{array}{c} X^{\prime }\\ Y^{\prime }\\ Z^{\prime } \end{array}\right]=\left[\begin{array}{ccc} \cos\theta_{z} & -\sin\theta_{z} & 0\\ \sin\theta_{z} & \cos\theta_{z} & 0\\ 0 & 0 & 1 \end{array}\right]\left[\begin{array}{c} X\\ Y\\ Z \end{array}\right]\;$$

The subsequent 2D boundary projection was used to identify the initial boundary for the aneurysm sac for each rotation. The boundary was then divided into two sets of nodes describing the left and right portions of each projection. Both the left and right boundary points were curve-fit using a trigonometric Fourier series. Initial testing of accuracy of the curve fitting used a range of terms (1 to 8 coefficients) for the Fourier series, and it was found that the following seven-term series (3) was sufficient when minimizing the root mean squared error:(3)$$y=a_{o}+\sum_{i=1}^{n}a_{i}\cos\left(iwx\right)+b_{i}\sin\left(iwx\right)$$

Here, *n* is the number of harmonics, *w* the fundamental frequency of the curve, *a_o_* the intercept constant, *a_i_* and *b_i_* are coefficients of the fitted Fourier series model.

The first and second derivative of each Fourier series function was then used to extract the start and end nodes of the aneurysm sac by choosing the local extrema from the second derivative as an inflection point of curvature change (Figure 2). The resulting aneurysm sac region of interest was defined by all nodes occupying the start and end boundaries.

Ad hoc analysis was run on an AMD (Advanced Micro Devices Inc., Santa Clara, CA) Ryzen Threadripper 2990WX 32-Core processor workstation with 128 gigabytes of DDR4 random access memory with a NVIDIA 2080Ti (NVIDIA Inc., Santa Clara, CA, USA) graphics processing unit using parallel pool computing.

### 2.6. Assessment of Computational Cost Dependence on Rotational Increment

In order to evaluate the computational cost associated with each rotational increment (*θ_z_* in Equation (1)), we determined the time necessary to isolate the aneurysm sac region as described above of each AAA studied here (*n* = 245). Rotational increments of 0.5°, 1°, 7.5°, 15°, 22.5°, 45°, and 90° were used and the time to process the entire aneurysm dataset was recorded.

### 2.7. Stress Calculations and Statistical Comparisons

The MWS and PWS were calculated for all cases and compared between the “entire aneurysm” (when using all of the nodes of the aneurysm) and the isolated “aneurysmal sac” (using only the subset of nodes that remained after sac isolation) using a paired *t*-test. The PWS was calculated and was defined as the 99th percentile wall stress found along the included geometry. MWS and PWS were compared between stable and unstable AAA using a Mann–Whitney U-Test (Wilcoxon rank-sum test).

## 3. Results

The mesh independence study (Figure 3A) revealed that the initial number of mesh elements (and mesh density) overestimated the peak wall stress until the solution converged after 2 million total elements. Therefore, all subsequent meshing targeted around 3 million elements when performing stress analysis. We found a strong dependence of computational time for sac isolation on the angular rotation increment (Figure 3B). The computational time exponentially increased for rotational increments less than 7.5°. By comparison, the 7.5° rotational increment took only 22 min (approximately 5 s per case) to process rather than 218 min for the 0.5° increment for all AAA cases, respectively.

Isolating the aneurysmal sac led to a statistically significant difference in MWS than that assessed for the entire aneurysm (10.6 ± 4.08 N/cm^2^ vs. 9.73 ± 3.29 N/cm^2^, *p* < 0.001). PWS was not significantly different in aneurysmal sac vs. entire aneurysm (20.8 ± 8.03 N/cm^2^ vs. 20.75 ± 7.51 N/cm^2^, *p* = 0.859).

Table 1 contains the mean (±standard deviation) MWS and PWS values for the sac-isolated and entire aneurysm for both groups of AAA, stable and unstable, and Figure 4 graphically compares these. The MWS for the aneurysmal sac alone for the unstable cohort was significantly higher than for the stable group (12.0 ± 3.63 N/cm^2^ vs. 10.5 ± 4.11 N/cm^2^, *p* = 0.0222). When comparing MWS of the entire aneurysm for these two groups, the statistical significance is lost (9.87 ± 3.25 N/cm^2^ vs. 9.71 ± 3.25 N/cm^2^, *p* = 0.509) (Figure 4A). A similar trend was observed for the PWS, wherein sac-isolation showed a statistically significant difference between unstable and stable groups (24.6 ± 7.06 N/cm^2^ vs. 20.5 ± 8.04 N/cm^2^; *p* = 0.003), while using the entire aneurysm did not (22.2 ± 8.07 N/cm^2^ vs. 20.6 ± 7.45 N/cm^2^; *p* = 0.157) (Figure 4B).

## 4. Discussion

In this paper, an objective and repeatable technique was developed to algorithmically isolate the aneurysm sac to assess AAA biomechanical metrics more precisely over what is the current standard of reporting these quantities for the entire aneurysm. Our results showed that sac isolation led to statistically significant differences in MWS for all AAA evaluated in this study. This highlights the importance of sac-isolation for MWS and the sensitivity of at least one biomechanical metric on the inclusion of extraneous regions outside the aneurysmal sac (e.g., non-dilated neck, branches, etc.). On the other hand, PWS was less sensitive to sac isolation across the AAA population because it is typically found in the aneurysm sac. Sac-isolation also reveals a statistically significant difference in both MWS and PWS between stable and unstable groups of AAA. Additionally, it was found that the sac isolation at a 7.5° rotational increment took only 22 min to process while yielding the same MWS results as smaller rotational increments. This is a significant reduction in computational time and power than using the smallest rotational increment of 0.5° (processing time of 218 min). Several reports that conducted FEA of AAA used MWS as a biomechanics metric since it provides guidance on whether an aneurysm, in a general sense, is at an elevated stress-state [12,14,15,16,17,19,25]. The 3D geometric reconstruction needed for computational analysis of AAAs can vary considerably. The results we presented here show the importance of standardizing a 3D reconstruction process, including, in particular, the region of interest over which biomechanical metrics are assessed. In cerebral aneurysms, Raghavan et al. implemented custom algorithms to quantify the morphology of ruptured and unruptured cerebral aneurysms [26] using a single cutting plane for the aneurysm sac. Later, an automated technique was developed by Larrabide et al., that also quantify morphological indices [27]. They reported an automated, user-independent, and repeatable isolation of cerebral aneurysm sac analogous to our approach in developing an approach to perform AAA sac isolation. Additionally, again in cerebral aneurysms, Piccinelli et al. published an automated neck plane detection algorithm to geometrically characterize sac boundaries [28]. Prior to these techniques, cerebral sac-isolation has been conducted manually.

To our knowledge, no previous reports have explored the impact of sac-isolation on biomechanical assessments of patient-specific AAA. However, studies have reported sac and neck markers of both growth and shrinkage of AAA [29,30,31,32,33]. However, there is not a detailed description of how the sac of the AAA sac is isolated from the neck. In previous work, our group investigated the wall stresses for idealized asymmetric bulges of AAA [18] and found that increasing asymmetry elevated wall stresses, albeit, where ILT was not present. Morphological based indices have been suggested to understand the transition from the proximal and distal ends to the bulge of the aneurysm sac [34]. However, axial slices in the XY-plane were used to identify this transition, not indicative of the actual diameter that is normal to the aneurysm geometry centerline. MWS is often overlooked, it is important to note that the population wall stresses in the sac region only could offer significant insight into the growth rate of a patient’s aneurysm. Other groups performing computational fluid dynamics analysis on aortic aneurysms have simplified complex 3D morphologies by using lower order models (0D and 1D) [35,36,37] to accurately produce similar results to the previous 3D analyses. The sac isolation we proposed slightly differs in that the 2D projected planes and 1D curve analysis serve as an objective and repeatable method to extract the superior and inferior aneurysm sac boundaries for reporting key biomechanical indices from stress analysis.

There are some limitations of this work that should be kept in mind. The aneurysm cohort analyzed here had geometries with varying degrees of complexity, though none involved the iliac arteries and renal arteries. It is plausible that sac isolation would not be appropriate for highly tortuous aneurysms. Further, this study focused solely on AAA that contained ILT. Future studies will compare aneurysm models without ILT incorporated into the finite element model and aneurysms that exhibit additional geometric features (such as involvement of iliac and visceral arteries) to improve guidance on the proposed wall stress percentile range.

Biomechanical analysis of AAA has been of great interest and has continued to explore metrics that can characterize the diseased state or predict adverse events, such as rupture. An objective algorithmic tool to consistently extract the aneurysm sac was developed to demonstrate the need to disregard non-aneurysmal regions. Using the sac-isolated regions in our analyses, we were able to detect differences in MWS between stable and unstable AAA cohorts, suggesting that AAAs that are going to exhibit symptoms or rupture will have elevated MWS compared with those that do not.

## 5. Conclusions

In this study, we present for the first time, an algorithmic method to isolate the aneurysm sac and demonstrated that it can be employed to detect biomechanical differences between clinically relevant groups of AAA that are not revealed using the entire aneurysm. While sac isolation has been performed in cerebral aneurysms, a methodological approach to do this for AAA has not been implemented until now. The approach for AAA vastly differs due to the sac being fusiform and asymmetric. This automated technique requires no user input and adds an additional 5 s for post-processing (when using a 7.5° rotational increment). Reporting sac isolated stress metrics in an algorithmically defined region can unify PWS and MWS across different research groups. Therefore, it is recommended that future image-based studies of AAA that report biomechanical indices adopt sac isolation to report calculated metrics.

## Figures and Tables

**Figure 1 bioengineering-09-00601-f001:**
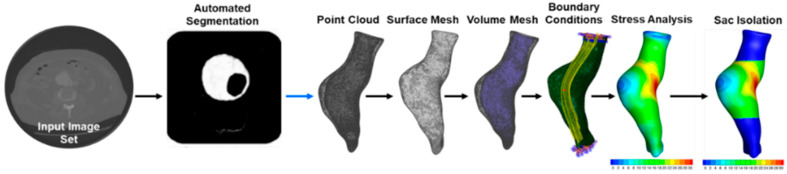
Image segmentation pipeline utilizing an automated segmentation technique based on a U-NET convolutional neural network. After the initial segmentation is accurately performed, the automated pipeline creates a finite element model by converting the point clouds into a mesh, assigning boundary conditions, performing FEA, and post-processing for aneurysm sac isolation.

**Figure 2 bioengineering-09-00601-f002:**
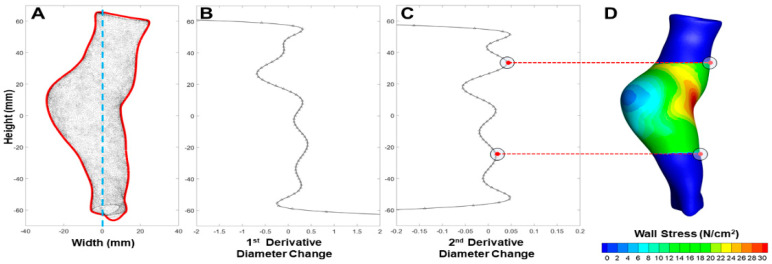
(**A**) 3D Scatter plot of an AAA model with the projected 2D boundary in red and a dividing blue line to split the left and right curves. A Fourier series (2) fit for the right projected boundary curve. (**B**) Subsequent first derivative of functionalized Fourier series. (**C**) The 2nd derivative of the Fourier series with the local extrema identified. (**D**) Sac isolated model in the same 2D projected view with local extrema assigning the superior and inferior boundaries on the right side of the aneurysm.

**Figure 3 bioengineering-09-00601-f003:**
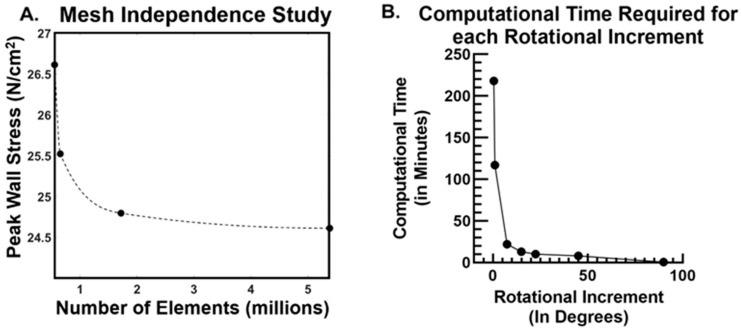
(**A**) Mesh independence study (data points were sampled) from ~600 k elements to 5.5 million elements. The points were spline fit to show a transition of a converging solution to report sac isolated peak wall stress values. (**B**) Total computational analysis time required to process all sac isolations (*n* = 245) plotted against the rotational increment. The increments evaluated were 0.5°, 1°, 7.5°, 15°, 22.5°, 45°, and 90°.

**Figure 4 bioengineering-09-00601-f004:**
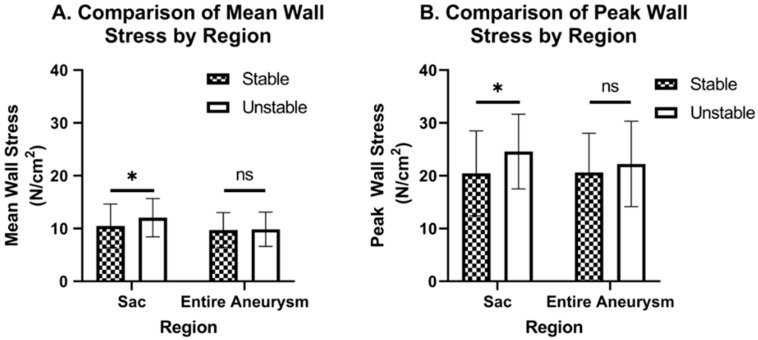
Comparison of MWS (**A**) and PWS (**B**) in sac and entire aneurysm regions, shows significant differences in stable versus unstable cases for the sac-isolated region but not for the entire aneurysm. The * symbol denotes a statistical difference (*p* < 0.05) and not significant is abbreviated as ‘ns’.

**Table 1 bioengineering-09-00601-t001:** MWS and PWS reported for both geometric regions analyzed and between stable and unstable AAA groups.

	Mean Wall Stress (N/cm^2^)	Peak Wall Stress (N/cm)
Stable Sac (*n* = 222)	10.5 ± 4.11	20.5 ± 8.04
Stable Entire aneurysm (*n* = 222)	9.71 ± 3.30	20.6 ± 7.45
Unstable Sac (*n* = 23)	12.0 ± 3.63	24.6 ± 7.06
Unstable entire aneurysm (*n* = 23)	9.87 ± 3.25	22.2 ± 8.07

## Data Availability

Not applicable.

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
