# Peer review of "An Objective and Repeatable Sac Isolation Technique for Comparing Biomechanical Metrics in Abdominal Aortic Aneurysms"

_bioengineering, 2022, doi:10.3390/bioengineering9110601_

Round 1

Reviewer 1 Report

The authors showed the utility of a novel sac-isolation algorithm by comparing peak/mean wall stress (PWS, MWS), with/without sac isolation, for AAA that were categorized as stable or unstable in 245 patient CT image sets.

The study is very interesting.

Methods are appropriately designed, results are relevant and well described, but Discussion section is too superficial and not focused on the main findings of the paper. It lacks also critical insight in the light of the current literature. It is also important to focus on the clinical application of your findings. Please rewrite and expand this section.

Reviewer 2 Report

The paper describes a topic of great interest, as the AAA is one of the most discussed pathology in the cardiovascular ambit. The paper presents an argument that is far from my competences, for this reason I feel not prepared to review the details of the materials and methods, that anyway seem well described and organized, as all the article. My question is what can be the usefullness from a clinical point of view? It could help if the values of MWS/PWS could be associated with particular characteristics of the aneurysm, as for example the aneurysm shape or the percentage of ILT.  In this way radiologists and surgeons would have an indication about which aneurysm is more at risk of rupture. Calculating for every patient MWS/PWS seems not possible in a clinical context; so it would be important to discover some characteristics that are often associated with higher values of MWS/PWS.

Reviewer 3 Report

1) Line 113, Page 3: CD8 is a linear hexahedral element but the authors by mistake wrote tetrahedral? So you can have either C3D4 or C3D10? I see the authors made the same mistake in their prior publication!

2) I checked the manuscript with iThenticate and seems like most of the method comes from their latest paper:

https://www.sciencedirect.com/science/article/pii/S2666496822000218?via%3Dihub

I would like to know the difference between this paper and the one that already published?

3) Line 107, Page 3: Fix the General?

4) Mesh density analysis? Please provide the results.

5) The geometry looks so smooth! What smoothing function and factor did you use? Gaussian?

6) The stress analysis has been done in Abaqus? the color bars look like workbench Ansys though?

7) Any mesh quality assessment?

8) Please provide the table for material properties?

9) How did you define anisotropy in the aneurysm? Fibers were defined?

This needs some explanation. 

10) Did you use explicit static mode? or implicit? 

11) The load and boundary conditions need to be specified?

I feel like the authors were too rush in the writing and explanation. Would appreciate if you proof read the manuscript before next submission.

Reviewer 4 Report

The paper written by the following Timothy Chung, Pete Gueldner, Trevor Kickliter, Nathan Liang, and David Vorp, entitled “An Objective and Repeatable Sac Isolation Technique for Comparing Biomechanical Metrics in Abdominal Aortic Aneurysms” presents an interesting numerical study directed into dedicated algorithm preparation for isolating the aneurysmal sac in patient-specific 3D reconstructed models for computational analysis of AAA.

Although the paper is interesting, I have some major concerns:

Title

The title reflects the results presented here.

Abstract

The abstract is lacking short material and methods description as well as an informative conclusion. It should be written in more details.

Introduction

In the introduction part Authors should add some overall information in  paragraph/paragraphs dedicated on 3d technique applied in medical diagnosis/treatment. The authors should consider justifying the motivation of this study with recent study and e.g. cite the paper listed below:

-         Spatial Configuration of Abdominal Aortic Aneurysm Analysis as a Useful Tool for the Estimation of Stent-Graft Migration; https://doi.org/10.3390/diagnostics10100737

-         A novel method for describing biomechanical properties of the aortic wall based on the three-dimensional fluid-structure interaction model. Interact Cardiovasc Thorac Surg. 2019 Feb 1;28(2):306-315. doi: 10.1093/icvts/ivy252.

Material and Methods

1.      Line 107: Start of the sentence is missing.

2.      There is no description of patients applied for the simulations. ? More details on patients recruitment, excursion and inclusion criteria should be added. It should be included in the manuscript.

3.      There is no information about the boundary conditions as well as initial conditions. It should be included in the manuscript.

4.      There is no information about the mesh independent test. It should be included in the manuscript.

Results

There is no information about the model validation. Authors did not compare their results with real patients. What is the accuracy of gathered trends. Could the Authors extrapolated their results into patients.

Discussion

Discussion part should be rewritten. Authors did not confront their results with current data on the topic.

Conclusions

Conclusions are too general and should be related to final results.

Round 2

Reviewer 1 Report

amended manuscript is acceptable. 

Reviewer 3 Report

All my comments have been addressed

Reviewer 4 Report

Unfortunately, Authors did not address most of my comments.

Introduction part is still missing the key information

Moreover, Authors did not explain the process of mesh independent test. There is no data presented concerning the specific parameters.

Still, there is no description of patients applied for the simulations. ? More details on patients recruitment, excursion and inclusion criteria should be added. It should be included in the manuscript
